# RNAi efficacy is enhanced by chronic dsRNA feeding in pollen beetle

Jonathan Willow [1,2✉], Liina Soonvald [1], Silva Sulg [1], Riina Kaasik [1], Ana Isabel Silva [3], Clauvis Nji Tizi Taning [2], Olivier Christiaens [2], Guy Smagghe [2] & Eve Veromann [1✉]

Double-stranded RNAs (dsRNAs) represent a promising class of biosafe insecticidal compounds. We examined the ability to induce RNA interference (RNAi) in the pollen beetle *Brassicogethes aeneus* via anther feeding, and compared short-term (3 d) to chronic (17 d) feeding of various concentrations of dsRNA targeting *αCOP* (dsαCOP). In short-term dsαCOP feeding, only the highest concentration resulted in significant reductions in *B. aeneus* survival; whereas in chronic dsαCOP feeding, all three concentrations resulted in significant mortality. Chronic dsαCOP feeding also resulted in significantly greater mortality compared to short-term feeding of equivalent dsαCOP concentrations. Our results have implications for the economics and development of dsRNA spray approaches for managing crop pests, in that multiple lower-concentration dsRNA spray treatments across crop growth stages may result in greater pest management efficacy, compared to single treatments using higher dsRNA concentrations. Furthermore, our results highlight the need for research into the development of RNAi cultivars for oilseed rape protection, given the enhanced RNAi efficacy resulting from chronic, compared to short-term, dsRNA feeding in *B. aeneus*.

[1] Institute of Agricultural and Environmental Sciences, Estonian University of Life Sciences, Tartu, Estonia. [2] Department of Plants and Crops, Faculty of Bioscience Engineering, Ghent University, Ghent, Belgium. [3] School of Mental Health and Neuroscience, Faculty of Health, Medicine and Life Sciences, Maastricht University, Maastricht, The Netherlands. ✉email: jonathan@emu.ee; eve.veromann@emu.ee

The pollen beetle *Brassicogethes aeneus* Fabricius (Coleoptera: Nitidulidae; synonym *Meligethes aeneus*) is a major pest of oilseed rape (*Brassica napus* L.) in Europe. Overwintered adult *B. aeneus* feed on pollen and nectar of blooming plants of various taxonomic families and later become monophagous on brassicaceous plants, where they obtain nutrients from reproductive buds and open flowers, followed by mating and subsequent oviposition into buds (usually 2−3 mm in length). Upon hatching from eggs, larvae feed on anthers within buds, followed by late first and early second instar larvae feeding in open flowers, the late second instar larvae eventually pupating in the soil under their host plant (reviewed in Mauchline et al.[1]).

As both the area of land used for and the number of people relying on crop production increases exponentially, the importance of achieving ecologically sustainable crop production also continues to grow. In order to achieve this, biologically safe strategies for managing crop pest populations are needed. A crop pest management strategy should consist of multiple approaches, together constituting one integrated pest management design. One widely suggested sustainable pest management approach is to enhance conservation biological control, based on the preservation or restoration of habitats and habitat features that provide food, alternative prey, shelter, overwintering sites and other natural resources to relevant natural enemies (i.e. predators, parasitoids) of crop pests; these conservation measures are ideally implemented at both local and regional scales[2−9]. The use of trap crops represents an additional pest management strategy, whereby an attractive companion crop diverts pests from the main crop, with the aim to reduce damage inflicted to the main crop; this strategy has shown promise for use in integrated *B. aeneus* management (reviewed in Skellern and Cook[10]). Another area of interest in oilseed rape protection, against *B. aeneus* and other insect pests, is the use of plant breeding techniques such as the exploitation of oilseed rape's natural variation, introgressive hybridisation with other brassicaceous species and the introduction of transgenes in the oilseed rape genome (reviewed by Hervé[11]). The application of insecticides can dramatically reduce yield losses in oilseed rape production. However, these compounds often kill non-target organisms, including economically beneficial insects that contribute to pest management[12−14]. In order to contribute to a biosafe outcome for non-target organisms, insecticidal compounds used should be as specific to the target pest as possible.

Post-transcriptional gene silencing via RNA interference (RNAi) represents a potential tool for use in an integrated and biosafe crop pest management design[15]. As RNAi occurs via the nucleotide sequence-specific mode of action of double-stranded RNA (dsRNA), this control measure may affect a desirably narrow range of species, taken on a case-by-case basis, depending on the design of target-specific dsRNAs. In brief, when a sequence homology exists between a small interfering RNA (siRNA) (20−24 nucleotides) processed in vivo from exogenous (e.g. ingested) dsRNA and an endogenous messenger RNA (mRNA), this homology allows the complementary region of endogenous mRNA to base-pair to the siRNA and become cleaved by the RNA-induced silencing complex ribonucleoprotein, preventing translation of the target mRNA. Two overarching strategies for inducing RNAi in crop pest populations have recently made significant progress in their development. One of these is host-induced gene silencing (HIGS) via the use of an RNAi cultivar, and the other is spray-induced gene silencing (SIGS) via sprayable dsRNA (reviewed in Christiaens et al.[16]). The prospects of the latter are further reviewed in Cagliari et al.[17]. HIGS has been shown to be an effective approach, for example, in controlling western corn rootworm (*Diabrotica virgifera virgifera* LeConte) via dietary exposure to transgenic maize (*Zea mays* L.) engineered

to produce dsRNA targeting the gene *v-ATPase A* in *D. virgifera virgifera*[18]. This approach has the benefit in that an RNAi cultivar constantly produces the target pest-specific dsRNA within the plant's tissues, chronically exposing the target pest to the sequence-specific insecticide, so long as the insect feeds on the transgenic crop. Indeed, the RNAi maize cultivar MON87411, expressing dsRNA targeting *Snf7* in *D. virgifera virgifera*, has been approved in several countries[19,20]. RNAi efficacy via a SIGS approach has been demonstrated in a greenhouse experiment where the foliar surface of 4-week-old potato (*Solanum tuberosum* L.) plants was treated with dsRNA targeting the gene *act* in Colorado potato beetle (*Leptinotarsa decemlineata* Say)[21]; as well as in a field trial where spraying of dsRNA targeting the *mesh* gene in the same species resulted in less leaf damage and greater *L. decemlineata* mortality, compared to the mortality observed on untreated plants[22]. This approach has the benefit of not requiring the biotechnology or time required for engineering an RNAi cultivar. However, two potential drawbacks include the possibility that exogenously applied dsRNA may not remain stable for long periods under natural outdoor conditions, and that successive applications may become necessary across stages of plant growth. The latter potential drawback is especially important to consider in the management of *B. aeneus* and other anthophilous species, as these acquire nutrients from flowering structures, which are in constant development and senescence, rather than leaves, which remain individually established on the growing plant for much longer periods. Thus, if a SIGS approach was to be put into practice within an integrated *B. aeneus* management framework for oilseed rape protection, the potential requirement of successive dsRNA spray applications must be considered.

Knorr et al.[23] first demonstrated oral RNAi and subsequent RNAi-induced mortality in *B. aeneus*, targeting several genes (e.g. *ncm*, *Rop*, *RpII140*, *dre4*) that were orthologous to RNAi-sensitive genes targeted in *D. virgifera virgifera* bioassays performed in the same study. An additional vital gene, *αCOP*, encodes the αCOP protein, a subunit of coatomer protein complex-I (COPI). COPI is involved in intracellular vesicular transport of proteins between the endoplasmic reticulum and Golgi apparatus; the transport of various other cellular cargo via its indirect interaction with the cytoskeletal motor protein dynein; and possibly the maintenance of protein distribution within the Golgi stack[24]. Furthermore, COPI is active in maintaining lipid homeostasis[25], and knockdown of COPI subunits inhibits protein and lipid accumulation at the cleavage furrow and reduces the number of microtubules at the central spindle, together resulting in cytokinesis failure[26]. Targeting *αCOP* expression, we recently demonstrated, under laboratory conditions, significant RNAi-induced mortality in *B. aeneus* via honey-solution feeding, simulating dsRNA-contaminated nectar[27]; as well as significant *αCOP* silencing via bud feeding, suggesting potential for developing an RNAi technique exploiting dsRNA-contaminated buds[28]. Besides carbohydrates from nectar and the lipid and protein constituents of buds, pollen beetles such as *B. aeneus* also consume pollen to acquire lipids and proteins, which helps *B. aeneus* in maintaining fitness at both the individual (e.g. energy storage) and population (e.g. gametogenesis) scale. Studies have suggested that while consumption of pollen positively influences *B. aeneus* survival and reproductive fitness, pollen is not critical for *B. aeneus* survival[29,30]. Nevertheless, *B. aeneus*'s consumption of pollen, together with the potential for SIGS or HIGS of *B. aeneus* populations via early-flowering trap crops, respectively treated with or bioengineered to produce dsRNA, makes it critical to examine RNAi efficacy via anther feeding in *B. aeneus*.

The aims of the present study were to confirm the ability to induce the RNAi effect via anther-based feeding of dsRNA, a field-relevant and so far unexamined dietary exposure route, and

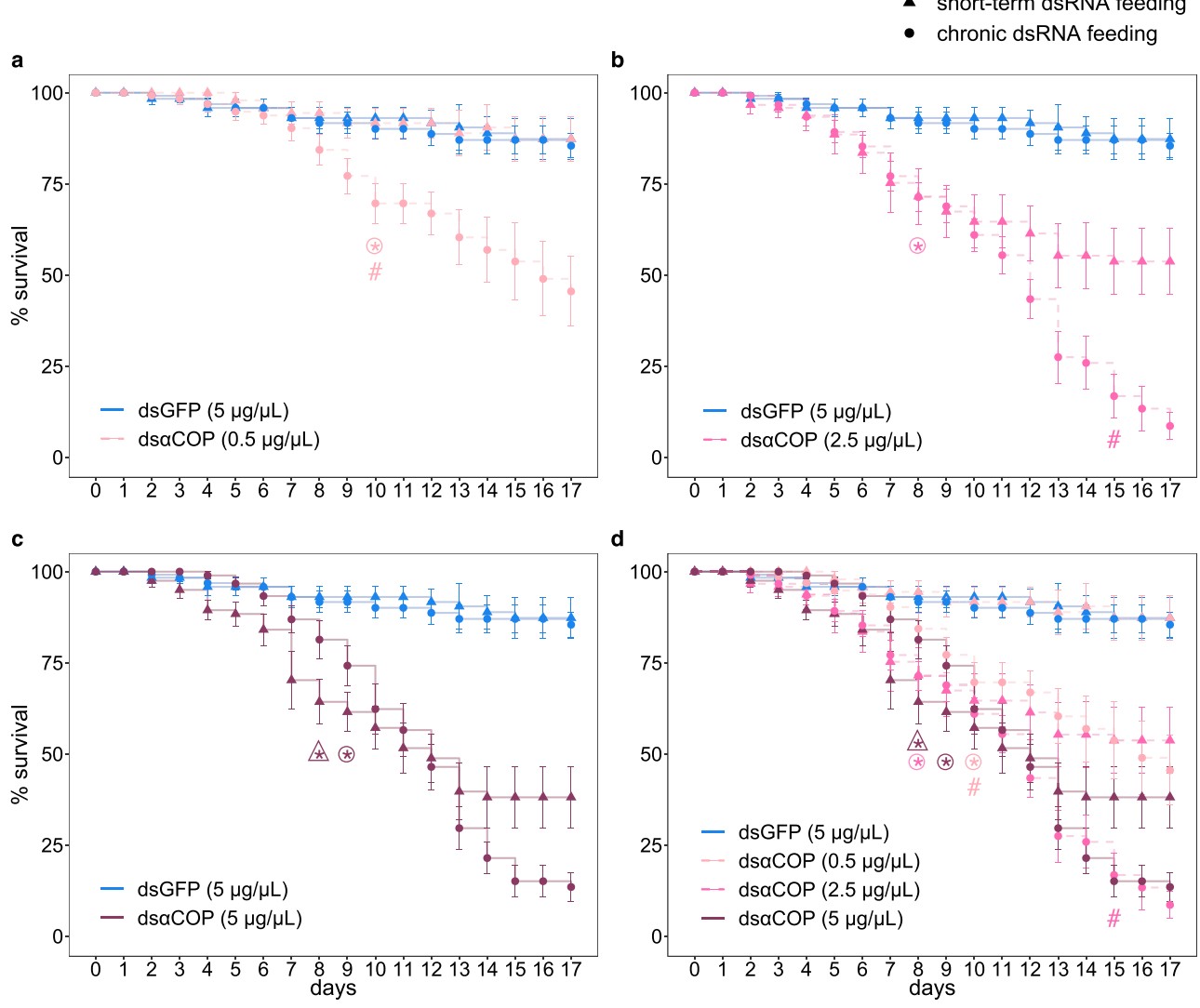

**Fig. 1 Survival (%) of *Brassicogethes aeneus* in each treatment in the RNAi assay, totalled over all three experimental replicates (starting *n* = 15 biologically independent cages of insects per treatment).** Survival curves show *B. aeneus* survival rates for short-term (3 days) and chronic (17 days) exposure to dsRNA treatments: dsαCOP at 0.5 µg/µL (**a**); dsαCOP at 2.5 µg/µL (**b**); dsαCOP at 5 µg/µL (**c**); and all dsαCOP treatments shown together (**d**). Asterisk (*) indicates significant difference ($p < 0.05$) in survival, compared to dsGFP 5 µg/µL (control) treatment. Colour of the asterisk indicates the corresponding dsRNA and concentration. Asterisk in a triangle indicates that the significance corresponds to short-term dsRNA feeding. Asterisk in a circle indicates that the significance corresponds to chronic dsRNA feeding. Hash symbol (#) indicates a significant difference ($p < 0.05$) in survival between short-term and chronic dsRNA feeding groups. Colour of hash symbol indicates the corresponding dsRNA and concentration. Asterisks and hash symbols are only used where values become and remain significant. Analysed using Kruskal–Wallis test, followed by Wilcoxon's rank-sum test with Bonferroni correction. Error bars: ±SEM.

to compare RNAi efficacy between short-term and chronic feeding of dsRNA targeting *B. aeneus* αCOP (hereafter dsαCOP), simulating two approaches to SIGS. We show that, for our low and medium dsαCOP concentrations, chronic dsαCOP feeding results in significantly greater mortality compared to short-term dsαCOP feeding, and that *αCOP* silencing was only significant in *B. aeneus* adults chronically fed dsαCOP-treated anthers. Considering the economics and development of a SIGS approach, our results highlight the potential for enhancing pest management efficacy via successive low-concentration treatments, compared to a single high-concentration treatment.

## Results

**Survival**. We observed significant reductions in *B. aeneus* survival as a result of consumption of dsαCOP-treated anthers for both

short-term (3 days) and chronic (daily for 17 days) dsRNA feeding (Fig. 1). With short-term dsRNA feeding, significant reductions in survival were observed starting at 8 days (64% survival) in the dsαCOP 5 µg/µL treatment (Fig. 1c), compared to the dsGFP control ($p = 0.007$) and dsαCOP 0.5 µg/µL ($p = 0.006$) treatments. Survival for short-term dsαCOP feeding at 5 µg/µL fell from 64% ($p = 0.007$, 8 days) to 39% ($p = 0.0096$, 13 days), afterwards remaining at 38% ($p = 0.005$) until the end of the experiment. Similarly, significant reductions in survival (65% survival, $p = 0.027$) were observed starting at 9 days in short-term dsαCOP feeding at 2.5 µg/µL (Fig. 1b), compared to dsGFP control, although this difference became statistically insignificant ($p = 0.08$) at 15 days; here, survival largely reached its lowest level at 13 days (53% survival, $p = 0.04$), afterwards remaining at 52% ($p = 0.08$) until the end of the experiment. When comparing the

dsαCOP 2.5 µg/µL treatment to the dsαCOP 0.5 µg/µL treatment (Fig. 1d), reductions in survival were marginally significant starting at 8 days ($p = 0.054$). Similar to the dsGFP control, short-term dsαCOP feeding at 0.5 µg/µL resulted in a total of 87% survival at 17 days (Fig. 1a). Thus, no difference in survival was observed between the dsαCOP 0.5 µg/µL and dsGFP treatments with regard to short-term dsRNA feeding.

With chronic dsRNA feeding, significant reductions in *B. aeneus* survival were observed starting at 8, 9 and 10 days for dsαCOP 2.5 µg/µL (72% survival, $p = 0.02$; Fig. 1b), dsαCOP 5 µg/µL (74% survival, $p = 0.03$; Fig. 1c) and dsαCOP 0.5 µg/µL (70% survival, $p = 0.036$; Fig. 1a) treatments, respectively. Survival for chronic dsαCOP feeding at 0.5 µg/µL continued to steadily fall to 46% ($p = 0.018$, 17 days), whereas survival for chronic dsαCOP feeding at both 2.5 and 5 µg/µL fell more rapidly, respectively reaching 26% ($p = 0.003$) and 30% ($p = 0.003$) at 13 days, and reaching their lowest levels at 8% ($p = 0.002$) and 13% ($p = 0.002$).

We also observed significant differences in *B. aeneus* survival when comparing short-term to chronic dsαCOP feeding. Starting at 10 days, chronic dsαCOP feeding at 0.5 µg/µL showed significantly reduced ($p = 0.04$) survival of *B. aeneus*, compared to short-term feeding of the same concentration (Fig. 1a); this difference became more significant further into the experiment (17 days, $p = 0.01$). Similarly, chronic dsαCOP feeding at 2.5 µg/µL showed significantly reduced ($p = 0.027$) survival of *B. aeneus*, compared to short-term feeding of the same concentration (Fig. 1b), starting at 15 days; this difference also became more significant further into the experiment (17 days, $p = 0.004$).

**Gene expression**. We observed contrasting results with respect to relative expression of *αCOP* between short-term and chronic dsRNA feeding groups (Fig. 2). Regarding short-term dsαCOP feeding, we observed a trend of reduced *αCOP* expression at 3 days. Here, we detected a 39% mean decrease in *αCOP* expression in the dsαCOP 0.5 µg/µL treatment ($t = 1.15$, d.f. = 2.79, $p = 0.34$), a 60% mean decrease in the dsαCOP 2.5 µg/µL treatment ($t = 1.95$, d.f. = 2.01, $p = 0.19$) and a 64% mean decrease in the dsαCOP 5 µg/µL treatment ($t = 1.85$, d.f. = 3.02,

$p = 0.16$), compared to the dsGFP control. At 6 days, our gene expression data showed no *αCOP* silencing (dsαCOP 0.5 µg/µL: $t = -0.8$, d.f. = 3.17, $p = 0.48$; dsαCOP 2.5 µg/µL: $t = 0.18$, d.f. = 4, $p = 0.87$; dsαCOP 5 µg/µL: $t = -0.06$, d.f. = 3.71, $p = 0.95$). At 12 days, we again observed no *αCOP* silencing (dsαCOP 0.5 µg/µL: $t = 0.25$, d.f. = 3.99, $p = 0.82$; dsαCOP 2.5 µg/µL: $t = -0.59$, d.f. = 3.18, $p = 0.6$; dsαCOP 5 µg/µL: $t = -1.14$, d.f. = 2.04, $p = 0.37$; Supplementary Fig. 1).

Regarding chronic dsαCOP feeding, differences in *αCOP* expression were more pronounced and statistically significant in some treatments, compared to the dsGFP control. At 3, 6 and 12 days, *αCOP* silencing was not observed in the dsαCOP 0.5 µg/µL treatment (3 days: $t = 0.61$, d.f. = 3.69, $p = 0.58$; 6 days: $t = 0.34$, d.f. = 2.95, $p = 0.75$; 12 days: $t = -0.33$, d.f. = 2.46, $p = 0.69$). Chronic dsαCOP feeding resulted in *αCOP* silencing in both the 2.5 and 5 µg/µL treatments at both 3 and 6 days. At 3 days, we observed a 63% mean decrease in *αCOP* expression in the dsαCOP 2.5 µg/µL treatment ($t = 4.45$, d.f. = 3.71, $p = 0.01$) and a 50% mean decrease in the dsαCOP 5 µg/µL treatment ($t = 2.81$, d.f. = 3.98, $p = 0.05$). At 6 days, we observed a 64% mean decrease in *αCOP* expression in the dsαCOP 2.5 µg/µL treatment ($t = 2.9$, d.f. = 3.97, $p = 0.049$) and a 64% mean decrease in the dsαCOP 5 µg/µL treatment ($t = 2.49$, d.f. = 3.93, $p = 0.069$). At 12 days after chronic dsαCOP feeding, no *αCOP* silencing was observed in either the dsαCOP 2.5 µg/µL treatment ($t = -0.18$, d.f. = 3.92, $p = 0.87$) or the dsαCOP 5 µg/µL treatment ($t = 0.81$, d.f. = 2.96, $p = 0.48$).

## Discussion

We aimed to induce RNAi via anther-based feeding of dsαCOP, and to compare RNAi efficacy between short-term and chronic dsαCOP feeding, simulating two SIGS approaches. Our data suggest that with chronic dsRNA feeding, reduced dsRNA concentrations can be applied in order to achieve a similar effect compared to that achieved from short-term exposure to higher dsRNA concentrations. Overall, these observations have important implications for the potential optimal practice and economics of a SIGS approach to managing crop pest populations. Specifically, our results suggest that while the management of

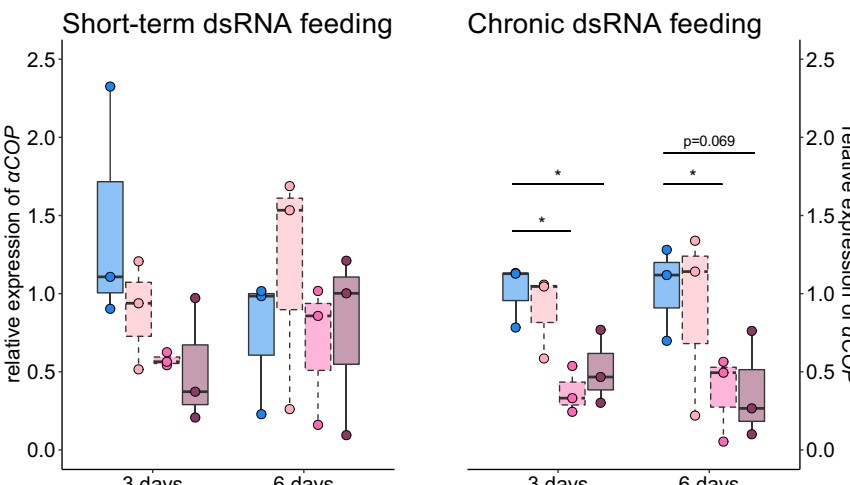

**Fig. 2 Results of gene expression analysis via quantitative polymerase chain reaction (qPCR) showing relative expression of *αCOP* in *Brassicogethes aeneus* at 3 and 6 days after the start of the experiment (*n* = 3 biologically independent qPCR samples per treatment and time point).** Target treatments (dsαCOP at 0.5, 2.5 and 5 µg/µL) are statistically compared to the dsGFP 5 µg/µL (control) treatment. Asterisk (*) indicates a significant difference ($p \leq 0.05$) between treatments. Analysed using Welch's *t* test.

*B. aeneus*, or of other crop pests, would likely benefit greatly from successive dsRNA spray treatments, this may also benefit the economics of dsRNA application, as lower concentrations may be suitable for an effective outcome. While we provide clear evidence to support this idea, semi- or small field experiments are necessary to further explore and confirm this RNAi approach. Furthermore, if SIGS was to be incorporated within a trap cropping approach, whereby adjacent trap crops were the target site for dsRNA spray treatments, this may represent a strategy for even greater efficiency of dsRNA use, as less total crop area would be treated.

We recently demonstrated significant *B. aeneus* mortality via 5 days of dsαCOP-treated honey-solution feeding, where significant mortality was first observed at 4 and 6 days in 3 and 1 µg/µL treatments, respectively, with survival continuing to steadily fall in the 1 µg/µL treatment, and falling more rapidly in the 3 µg/µL treatment[27]. The present study, which shows significant gene silencing-induced mortality in *B. aeneus* via consumption of anthers of dsRNA-treated flowers, provides evidence that suggests the potential for using an RNAi-based approach within *B. aeneus* management. Treating anthers is indeed highly field relevant when the target organism is an anthophilous species. However, compared to honey-solution containing similar concentrations of dsRNA, the amount of dsRNA being consumed when applied to anthers is likely far lower. As *B. aeneus* consumes large amounts of pollen via anther feeding, this difference in total dsRNA concentration between *B. aeneus*'s nutrient sources is of vital consideration for any potential SIGS approach to *B. aeneus* management. Similarly, we recently observed only 12 and 16% *B. aeneus* adult mortality at 10 and 15 days, respectively, after short-term (3 days) feeding on oilseed rape buds exogenously treated with dsαCOP at 5 µg/µL[28], compared to the present study in which we observed 43% (10 days) and 66% (15 days) mortality after short-term feeding of anthers treated with dsαCOP at 5 µg/µL. This difference in RNAi efficacy between bud- and anther-based dsRNA feeding is similar to that between honey-solution- and anther-based dsRNA feeding, in that *B. aeneus* adults chew through and consume bud epithelial tissue mostly to acquire nutrients from the anthers within the bud, thereby being orally exposed to a smaller amount of exogenously applied dsRNA compared to when feeding on dsRNA-treated anthers.

While dsRNA applications are unlikely to result in target species mortality as quickly as with some other (e.g. neurotoxic) insecticides, the benefit of using a dsRNA-based insecticide is its associated biosafety to non-target species due to the unique mode of action of dsRNA. There indeed also remains the potential for enhancing efficacy and speed-to-effect of dsRNA via co-formulants (e.g. nanoparticles) that can improve the efficiency of dsRNA uptake and RNAi[31,32]. Improving the efficacy of this technology will be a critical step to more fully realising the potential for using a SIGS approach in managing *B. aeneus* and other anthophilous pest species. The present study, together with our previous RNAi studies on *B. aeneus*[27,28], suggests that *B. aeneus*'s sensitivity to oral RNAi, via field-relevant routes of exposure, is relatively moderate compared to some other crop pests. Indeed, several coleopteran taxa have demonstrated sensitivity to dsRNA via feeding. For example, in addition to *L. decemlineata* and *D. virgifera virgifera*, on which much RNAi work has been done[18,21–23,33–36], robust sensitivity to oral RNAi has been observed in the Sri Lanka weevil (*Myllocerus undecimpustulatus undatus* Marshall)[37], the sweetpotato weevils *Cylas brunneus* Fabricius and *Cylas puncticollis* Boheman[38,39] and the lady beetle *Henosepilachna vigintioctopunctata* Fabricius[40].

Pollen beetle larvae begin development within the flower bud, feeding on anthers, and are typically in their late first or early second instar when oilseed rape flowers bloom; at this point, *B.*

*aeneus* larvae proceed to feed upon the anthers of open flowers of oilseed rape (reviewed in Mauchline et al.[1]). Therefore, it is plausible that dsRNA applications during flowering could target both larval and adult *B. aeneus* stages simultaneously. Studies examining the potential for anther feeding-induced RNAi in *B. aeneus* larvae would greatly enhance our understanding of the potential for using an RNAi approach in *B. aeneus* management.

While the present study used exogenously applied dsRNA to bring about gene silencing-induced mortality, our results raise the question of whether HIGS or SIGS represents the most optimal and effective RNAi approach to agricultural pest management. Current restrictions prevent the agricultural use of RNAi cultivars within European Union countries. However, this could change as our experience with this technology and our understanding of its impacts increases. RNAi risk assessment is a concept and practice that is under constant refinement, and is expected to provide evidence of the biosafety of RNAi cultivars, naturally on a case-by-case basis[20]. Based on our results, it is conceivable that a HIGS approach, exploiting the continuous production of target-specific dsRNA, could represent the optimal approach to RNAi-based management of *B. aeneus*, and possibly other crop pest species. However, the development of RNAi cultivars, and experiments simulating both HIGS and SIGS approaches, are necessary steps to more fully understand the practical differences between these approaches to RNAi-based crop pest management. In the context of *B. aeneus* and other anthophilous species, it remains critical to consider the constant development and senescence of flowers within the crop and the implications this may have for a SIGS approach, especially with respect to the potential requirement of successive dsRNA spray applications over the flowering season.

Ecologically sustainable agricultural pest management is required in order to attain ecologically sustainable crop production. Insecticides based on dsRNA represent a potentially species-specific complement to other biosafe measures (e.g. conservation biocontrol) for managing agricultural pests due to the unique mode of action of dsRNA. Our work demonstrates major differences between short-term and chronic feeding of target-specific dsRNA with regard to both gene silencing and gene silencing-induced mortality in *B. aeneus*, and suggests that similar differences may be important factors in other crop pest species. Our results also provide further evidence of the potential for RNAi-based management of *B. aeneus*, particularly via a SIGS approach utilising appropriately timed spray applications during the oilseed rape flowering period; but also applies to the ongoing conversation regarding the use of HIGS vs SIGS in crop pest management. Focal points critical for progress here include determining the duration at which exogenously applied dsRNA remains viable on the anthers of flowering crops as well as the optimal duration of exposure to dsRNA-treated anthers, taking into account the potential length of time between blooming and senescence of flowers within a given crop. It is also critical to determine the potential for management of *B. aeneus* larvae via consumption of dsRNA-treated anthers, as well as the overall feasibility of adopting a SIGS approach for controlling anthophilous pest species, in the context of potential requirements for successive dsRNA spray applications. Lastly, the development of RNAi cultivars for use in experiments must be considered in order to examine the potential for RNAi-based management of crop pests via HIGS, and simulate this against different SIGS approaches.

## Methods
**Double-stranded RNAs**. A selected 222 base pair (bp) region from the *B. aeneus* *αCOP* coding sequence, and a 455 bp sequence from the gene *gfp* (Supplementary Table 1), were used as the basis for in vitro synthesis of corresponding dsRNA products by AgroRNA (Genolution, Seoul, South Korea). These products contained

dsRNAs with sequences complementary to the genes *gfp* and *αCOP* and are, respectively, referred to as dsGFP (control) and dsαCOP. Both dsRNA products were shipped in distilled water (dH₂O) at ambient temperature and kept at 5±1 °C once received. The absence of nucleic contaminants in both the dsGFP and dsαCOP stocks was confirmed via gel electrophoresis.

**Insects and flowers**. Pollen beetles and oilseed rape flowers were both collected fresh from an untreated organic oilseed rape crop (58.36377°N, 26.66145°E) in the village of Õssu, Tartu County, Estonia. Beetles were kept in ventilated plastic containers and allowed to feed ad libitum on oilseed rape flowers. All pollen beetles were identified via Kirk-Spriggs[41], and only *B. aeneus* were used in experiments.

**Experimental setup**. *Brassicogethes aeneus* adults were placed into transparent, polystyrene, ventilated insect breeding dishes (diameter 10 cm × height 4 cm) (SPL Life Sciences, Gyeonggi-do, South Korea), hereafter referred to as cages. Eight fast moving beetles (used as a proxy for good health) were selected at random and introduced to each cage. Treatments were provided as ad libitum access to dsRNA-treated anthers of oilseed rape flowers, where petals were removed from flowers, and anthers were soaked in the treatment solution for 15 s, and subsequently allowed to air dry. All treated anthers were treated on the day of their provision. All anthers provided were dehisced, and thus pollen grains were freely available to the insects. Treatment solutions contained a given amount of dsRNA diluted in dH₂O and a constant concentration (180 ppm) of the surfactant Triton X-100 (Fisher Bioreagents), and were vortexed for 10 s prior to soaking anthers. There were eight treatments, including dsαCOP at 0.5, 2.5 and 5 µg/µL and dsGFP at 5 µg/µL, each provided for 3 days to one group (receiving untreated anthers after 3 days) and with another group receiving daily (17 days) treatment (hereafter respectively referred to as short-term and chronic dsRNA feeding). Fresh anthers were provided every 24 ± 1 h. Once provisioned with dsRNA-treated or untreated anthers, beetles were maintained in a climate chamber (Sanyo MLR-351H, Osaka, Japan) at 20 °C, 60% relative humidity and 16:8 h light–dark cycle. Each cage was additionally provisioned with a small piece of dental cotton roll saturated with dH₂O to provide drinking water for beetles. The experiment was replicated three times over 3 consecutive days, each time allocating five cages per treatment (starting *n* = 15 cages; 120 insects per treatment). Experimental setup is illustrated in Fig. 3.

Each experimental replicate lasted for 17 days. For each replicate, after 1 day, any dead beetles were removed from the experiment, as at this time the mortality could not have been due to RNAi, but rather likely due to stress resulting from manipulations and changing conditions; these mortalities after 1 day were few, and were accounted for in the statistical analysis. Survival was monitored every 24 ± 1 h, and dead insects were removed from cages daily.

Relative gene expression analysis was performed for all treatments via quantitative polymerase chain reaction (qPCR), represented by the time points 3, 6 and 12 days. At 3 and 6 days after the start of each experimental replicate, one cage was randomly removed from each treatment (min 6, max 8 beetles per sample; qPCR sample *n* = 3 cages; leaving *n* = 12 and 9 after 3 and 6 days, respectively, for survival analysis). At 12 days, for each experimental replicate, one beetle was removed from each remaining cage and used for qPCR analysis (three beetles were pooled per replicate; qPCR sample *n* = 3). The removal of beetles for qPCR was accounted for in the statistical analysis. Beetles used for qPCR were immediately placed in their respective Eppendorf tubes and homogenised using a sterilised plastic pestle designed for Eppendorf tubes in 600 µL of RLT buffer (by adding 10 µL of β-mercaptoethanol) and stored at −80 °C until analysis. Total RNA was extracted using the RNeasy Mini Kit (Qiagen, Venlo, The Netherlands); and RNA concentration was quantified, and purity was assessed, using a Nanodrop spectrophotometer (Thermo Scientific, Wilmington, USA). The purity was further verified via gel electrophoresis. Genomic DNA was removed using the Turbo DNA-Free Kit (Invitrogen, Carlsbad, USA), following the manufacturer's protocol. The complementary DNA (cDNA) was reverse transcribed from 1 µg of total RNA using the FIREScript RT cDNA Synthesis Kit (Solis BioDyne, Tartu, Estonia), and qPCR was performed in the QuantStudio 5 Real-Time PCR System (Applied Biosciences, Foster City, USA). The reaction mixture included 4 µL of 5×HOT FIREPol EvaGreen qPCR Supermix (Solis BioDyne), 0.5 µL of 10 µM forward primer (Microsynth; Supplementary Table 2), 0.5 µL of 10 µM reverse primer (Microsynth; Supplementary Table 2), 14 µL of PCR-grade water and 500 ng of cDNA, to a total volume of 20 µL. Amplification conditions were as follows: 15 min at 95 °C, 40 cycles of 15 s at 95 °C and 1 min at 58 °C, followed by a melting-curve analysis with a temperature range of 60–95 °C. The reactions were set up in 384-well PCR plates in triplicate. Normalisation of data was performed using the two housekeeping genes *rps3* and *act*. Primer amplification efficiencies were determined from a cDNA dilution series. Primer sequences and amplification efficiencies are shown in Supplementary Table 2. Relative gene expression values were calculated using the $2^{-\Delta\Delta Ct}$ method. A no-template control and a no-reverse transcriptase control were both included in the assay.

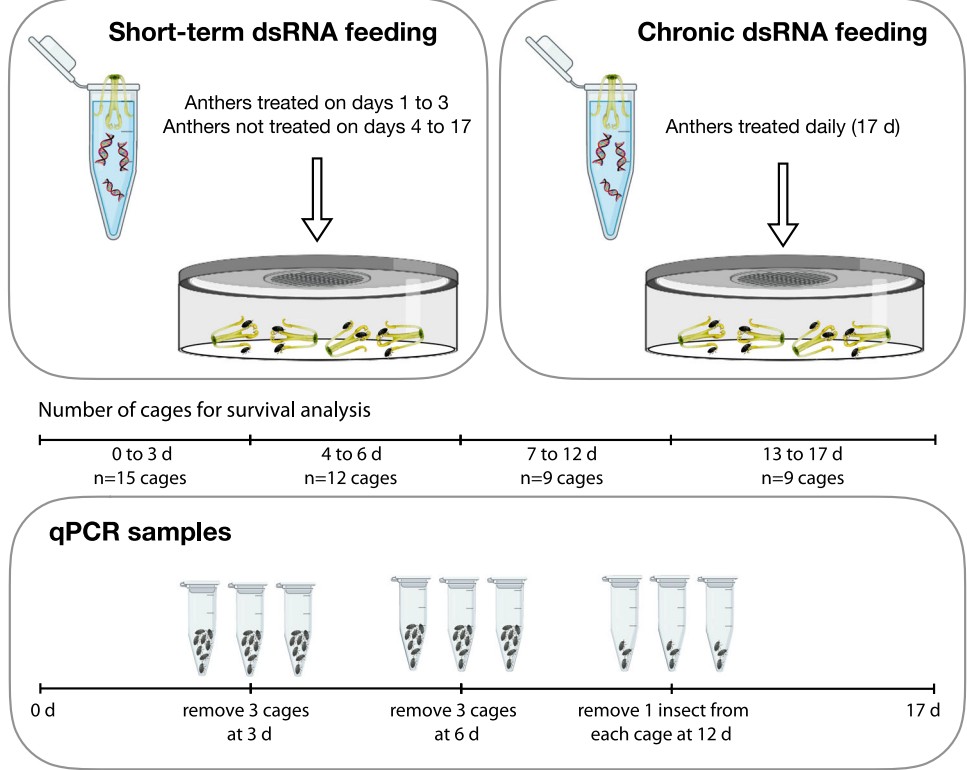

**Fig. 3 Experimental setup for each treatment (dsαCOP at 0.5, 2.5 and 5 µg/µL, and dsGFP at 5 µg/µL) for both short-term (3 days) and chronic (17 days) dsRNA feeding in *Brassicogethes aeneus* RNAi assays.** Here, we monitored *B. aeneus* survival and corresponding *αCOP* expression in specimens. For survival analysis, starting *n* = 15 cages per treatment, each cage with eight insects. For both short-term and chronic dsRNA feeding, three cages (*n* = 3) were removed for qPCR analysis at both 3 and 6 days, and one insect was removed from each of the nine remaining cages at 12 days for each treatment.

**Statistics and reproducibility**. Treatment comparisons taken into consideration are listed in Supplementary Table 3. Regarding survival analysis, for dsGFP and all three dsαCOP concentrations, comparisons between short-term and chronic exposure were statistically assessed. In addition, comparisons in survival were made between dsGFP and all three dsαCOP concentrations, as well as between dsαCOP concentrations, within both short-term and chronic exposure groups. When comparing different dsRNAs or concentrations, comparisons in survival were only made between treatment groups that were given the same duration of exposure to dsRNA. Regarding gene expression analysis, comparisons were made between dsGFP and all three dsαCOP concentrations within both short-term and chronic exposure groups. For survival analysis, homogeneity of variance and normality of data distributions were determined using the Levene and Shapiro–Wilk tests, respectively. Since the data were overall not normally distributed, the Kruskal–Wallis test was used as a nonparametric alternative to analysis of variance; this was followed by the Wilcoxon's rank-sum test, with Bonferroni correction, for post hoc pairwise comparisons. For gene expression analysis, comparisons were made using Welch's $t$ test. All statistical analyses were performed in R version 3.6.3 (R Foundation for Statistical Computing, Vienna, Austria).

The experiment was replicated three times, each experiment consisting of five cages (total $n = 15$ cages) per treatment. At both 3 and 6 days after each experiment, one cage of beetles was removed from the bioassay and was used to analyse relative gene expression (total $n = 3$ qPCR samples) for each treatment. Thus, for survival analysis, the sample size per treatment was reduced from $n = 15$ to $n = 12$ at 3 days, and from $n = 12$ to $n = 9$ at 6 days.

**Reporting summary**. Further information on research design is available in the Nature Research Reporting Summary linked to this article.

## Data availability
All data are available upon request from the corresponding author J.W.

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

## Acknowledgements

We thank the farm owner whose land was used for collecting pollen beetles and oilseed rape flowers. Jonathan Willow is the recipient of a Ph.D. grant from the European Social Fund for Doctoral Students, as well as the Internationalisation Programme DoRa (carried out by the Archimedes Foundation). O.C. is a recipient of a postdoctoral fellowship from the Research Foundation—Flanders (FWO-Vlaanderen). The authors acknowledge financial support from the Institutional Research Funding project no. IUT36-2 of the Estonian Research Council, the Institutional Research Funding project no. PRG1056 of the Estonian research council, the European ERA-NET C-IPM project "IPM4Meligethes" (project no. 3G0H0416), the European Union's European Regional Development Fund (Estonian University of Life Sciences ASTRA project "Value-chain based bioeconomy"), the Special Research Fund (BOF) of Ghent University and the Research Foundation—Flanders (FWO-Vlaanderen).

## Author contributions

J.W. conceived the experiment, designed methods, performed the experiment, supervised experiment, analysed and visualised data, validated analyses, wrote the original draft and suggested edits to revised manuscript versions. L.S. performed the experiment and suggested edits to revised manuscript versions. S.S. performed the experiment and suggested edits to revised manuscript versions. R.K. performed the experiment. A.I.S. analysed and visualised the data, validated analyses and suggested edits to revised manuscript versions. C.N.T.T. designed methods, validated analyses and suggested edits to revised manuscript versions. O.C. designed methods, validated analyses and suggested edits to revised manuscript versions. G.S. conceived experiment, validated analyses and suggested edits to revised manuscript versions. E.V. conceived the experiment, designed methods, provided resources, supervised experiment, validated analyses and suggested edits to revised manuscript versions. All authors read and approved the final manuscript.

## Competing interests

The authors declare no competing interests.
