## [Peer Review File · Communications Biology]

Reviewers' comments:

Reviewer #1 (Remarks to the Author):

This manuscript details work that will contribute to biosafe methods of crop protection for arable pests using the pollen beetle (*Brassicogethes aeneus*) in oilseed rape as a model. The authors tested the potential of double stranded RNA (dsRNA) targeted against the *B. aeneus* α -COP protein to produce RNAi interference and pest mortality via anther feeding. The dsRNA was presented at three concentrations and short-term exposure (3d) was compared to chronic exposure (17d) to simulate single or repeated applications (i.e. simulating two spray-induced silencing approaches) and relative gene expression was checked at 3, 6 and 12d via qPCR to determine gene silencing. Significant reductions in *B. aeneus* survival was observed in both short-term and chronic exposure. In the long-term exposure treatment, significantly fewer beetles survived at all concentrations compared to the control but in the short-term exposure treatment only the highest concentration resulted in reduced survival; chronic exposure also resulted in greater mortality than short term exposure at the same lower two concentrations. Only chronic dsRNA resulted in significantly reduced gene expression compared with the control at the two higher concentrations. The authors discuss the results in terms of use cases in spray-induced silencing strategies (SIGS) vs host-induced silencing strategies (HIGS), concluding that in a SIGS approach, treating anthers with dsRNA could be an effective pest management strategy and although a single application of high-dose dsRNA could be effective (offering only short-term exposure), successive applications could be more cost effective as lower doses could be used to achieve chronic exposure; a host-induced gene-silencing approach (HIGS, that would also confer chronic exposure) would also be potentially valuable.

The concept of RNAi is still relatively new and offers exciting possibilities for targeted control of pests such as *B. aeneus*. The practical work in this study is novel has been very well conducted, statistics are appropriate, and the manuscript is generally well written (see minor comments below). It is an exciting development in the quest for biosafe methods for pest management in oilseed rape which will interest many researchers and practitioners interested in crop protection. The discussion did not mention a couple of fairly obvious flaws in logic regarding the use-case for *B. aeneus*. Addressing these could make the manuscript even stronger.

My greatest concern is that this use case as presented would involve treating a flowering crop, probably repeatedly, to target pollen beetles feeding on open flowers. This by itself would only help to reduce the population of pollen beetles for the next season and would not help to reduce economic damage to the crop, which is caused mainly by feeding damage to closed buds before the flowering phase; during the flowering phase the beetles are not considered a pest. I feel this approach may not be very popular with farmers! The dsRNA technology would therefore be better sprayed at the bud stage, which is the damage-susceptible stage to this pest, to prevent feeding damage on buds - as in the authors' previous work which is not mentioned (Willow et al. 2020 First Evidence of Bud Feeding-Induced RNAi in a Crop Pest via Exogenous Application of dsRNA. *Insects* 11.11:769). This needs to be acknowledged. However I can envisage a practical and successful use case for the SIGS or HIGS approach via feeding in open flowers as part of a more integrated trap cropping strategy (i.e. where an early flowering trap crop is treated (SIGS) or used (HIGS) to prevent infestation and damage of the main cash crop during its damage-susceptible bud stage). This would have the added advantage that only a small area is treated and so would be economically more viable than treating the whole crop.

Secondly, this work follows a previous study by the authors presenting proof of concept of RNAi via nectar feeding (Willow et al submitted). However, the authors argue in this manuscript that it is critical to examine RNAi efficacy via anther feeding as *B. aeneus* is anthophilous (L92). Pollen is not critical to the survival of pollen beetle (see Cook et al 2004 Do pollen beetles need pollen? The effect of pollen on oviposition, survival, and development of a flower-feeding herbivore. *Ecological Entomology* 29 (2):164-173; and Corda et al (2018) Impact of flower rewards on phytophagous insects: importance of pollen and nectar for the development of the pollen beetle (*Brassicogethes*

aeneus). *Arthropod-Plant Interactions* 12(6):779-785). This phrasing therefore needs to be modified but moreover, this raises the issue that if pollen beetles can detect treated anthers and prefer untreated material, they could switch to feeding on other floral parts including nectar, which would decrease efficacy of RNAi via anther feeding. This supports the potential inclusion of nectar-mediated RNAi technology in the crop protection strategy as suggested in their previous work (Willow et al submitted). I feel, however, that if pollen beetles can detect and avoid treated anthers only a HIGS approach could be successful as untreated anthers would be common in a SIGS approach with new flowers appearing daily - repeated spraying could therefore be ineffective and/or impractical (and would also increase the carbon footprint of the crop production strategy).

Minor comments:

The manuscript would benefit from certain terms being better defined to help the non-specialist reader:

1. How does double stranded RNA work (briefly) in terms of producing RNA interference effect? (Introduction)
2. what is α COP (ds α COP) (Abstract & introduction) and what does the α COP protein do to make it vital? (L88)
3. bp should be written in full for first use in Methods section (L224)

Some clarifications to the M&M would be helpful:

1. Abstract - duration of short-term exposure (3d) given but not term of chronic exposure (17d) ;
2. L239 Were anthers from treated flowers of various ages or from flowers of a standard age? was the pollen dehisced or not or a mixture of both?
3. L241 How long was treatment vortexed for?
4. L103 & 243 – It is unclear whether anthers for the short term treatment were provided with treated anthers which were left for 3d or if anthers were changed daily as with chronic treatment anthers (Fig 1 suggest the latter) -needs to be made clear in the text as well as the duration of the chronic exposure (= experimental duration 17d) – this should come sooner than L250.
5. Why was experimental duration 17d and not, say 14 or 21 days?

Grammar / spelling needs attention in places.

1. In the first line of the abstract 'represents the most biosafe insecticidal compounds' - represent should be singular? Also, there are two 'biosafe' words in the same sentence. Last sentence of the Abstract also needs attention (two approaches, clunky split);
L64-67 sentence is clunky with use of two semicolons;
L83 as these acquire;
L89 (Willow et al. In Press) should come at the end of the sentence;
L81 two uses of 'successive' in this sentence;
L83 acquire ; L96 & 105 short term not shortterm; L106 set-up not setup;
L231 as pollen is not critical for survival as mentioned above, please also delete 'on pollen of'
L232 crop not field - the field contains the crop!; Eight fast moving beetles (used as a proxy...) were selected at random and introduced... ;
L247 wettened? Past tense of wet is wetted – was it saturated or moistened? these would possibly be better terms;
L155 no comma;
L159 replace 'as that which would' to 'compared to that achieved' L160 a higher concentration or higher concentrations;
L164 evidence to support (not of) this idea;
L184 anthophilous;
L186 delete larvae after instar;
L188 insert stages before simultaneously;
L192 rephrase 'conceivably touch upon the concept of HIGS vs SIGS are the most optimal' – it's a

question not a concept?

L200 development of an RNAi cultivar or development of RNAi cultivars ;
L209 target specific.

Several references are needed:

1. L66 the first strategy needs a reference?;
2. L230 reference for use of gel electrophoresis for this purpose?;
3. L234 reference for species determination of *B. aeneus* from other related species;
4. L255 reference for relative gene expression analysis via qPCR? 5. When names are cited in the text these seem not to be numbered as with other citations e.g. Mauchline et al 2018 (L51; #1) , Cagliari et al 2018 (L67; #14) are missing citation numbers ; Willow in press should be ref #20 and should be cited (L89, 169)?

Tables & Figures. Legends should be written in full such that all necessary information is presented to the reader to negate reference to the text. This related to all Figures and tables: 1. Table 1 - treatment comparisons are the same in each category, so I'm not sure this is really needed - could explain in the text or move to Supplementary material? ;
2. Fig 1 should describe what the 'experiment' was for (note set-up, not setup);
3. Figure 2. Describe 'each experiment' i.e. what were the treatments, rates, & brief description of what the test was. Is the % survival totalled over 3 reps or is it the mean % survival over 3 reps - what do the error bars represent SE mean or SE proportion? Key is difficult to interpret - different treatments (dsGFP and dsCOP) would be better just a coloured line with no circle as begs question about short-term feeding treatments (triangles). L403 Asterisk. Figs A-C are not strictly necessary - could Fig D be drawn on a finer scale to enable clearer separation between the different treatments?;
4. Fig 3 needs clearer description of the experiment i.e. results of gene expression to measure success of gene silencing, qPCR should be given in full legend, what are the units on the Y axis, explain what at 3d and 6d means i.e. from the start of the experiment/after feeding for 3d/6d on treated anthers.

I enjoyed reading this manuscript!
Sam Cook

Reviewer #2 (Remarks to the Author):

Review of "RNAi efficacy is enhanced by chronic, compared to short-term, dsRNA feeding in a crop pest" by Jonathan Willow and colleagues, for publication in Communications Biology.

The authors demonstrate an increased RNAi efficacy in the pollen beetle *Brassicoglyphus aeneus* by constant feeding of dsRNA via a new feeding bioassay. Anthers of the oilseed rape *Brassica napus* were treated with dsRNA targeting the coatamer subunit alpha (α COP) and fed to *B. aeneus* in three concentrations. Either short-termed for 3 days or continuously for the duration of the assay. The latter resulted in significantly greater mortality.

The manuscript is well written, especially the detailed method section is of great value for researchers that would like to adapt the established anther feeding assay. Parts of the paper, however, might need some rewriting and additional information.

Please see comments below:

Title

I recommend to include the pest name *Brassicoglyphus aeneus* and or pollen beetle, as this might increase the visibility of the manuscript

Abstract

- Line 29: "potentially represents" exchange with e.g. "represents a promising ..."
- Line 30: "...biosafe strategy for integrated management of the pollen beetle *Brassicoglyphus aeneus* is greatly needed ..." Why?
- Line 32: "...ability to produce the RNA interference (RNAi) effect..." I suggest to use a different wording e.g. "induce" or "initiate" RNAi effects

- Line 39-41: "Our results have implications for... as well as the need for research into the development and potential future use of host-induced gene silencing approaches." Is there something missing in this sentence?
- Line 39: "Our results have implications for the development of RNAi strategies ..." Please state which implications.
- Line 41: "the need for research into the development and potential future use of host-induced gene silencing approaches" Why?
- Line 66: "this control measure is potentially species-specific" Reference?

Keywords

- Exchange old name "Meligethes aeneus" with "Brassicogethes aeneus"

Introduction

- Line 53-62: This section seems to me a bit unstructured. Starting with "In order to achieve ecologically sustainable crop production...", but not mentioning why we have to achieve this. Next, the authors' states that an ecologically sustainable crop production should consists of "multiple approaches", but list only one approach: "enhancing biological control". This followed by the application of insecticide use. The authors should consider rewriting this section.
- Line 89: The authors should consider citing the work of Knorr and colleges (2018) that showed functional RNAi in field collections of *B. aeneus*, as well as some information/references about the selected RNAi target gene
- Line 95: exchange "produce" with "induce"; add "of dsRNA" after "anther feeding"
- Line 97-99: delete "Using gfp specific dsRNA (dsGFP) as a control" (Material & Methods)
- Line 98-100: At the end of the introduction you usually give a brief description of important results, and not the authors' expectations.

Results

- Line 103-107: Belongs to Material and Methods

Discussion

- Line 155-157: too much details, belongs to Material and Methods. Instead the authors should start with re-stating the research problem they wanted to investigate
- Line 159: "similar effect as that which would be achieved from" sounds odd
- Line 161-164: I can't follow the statement of this sentence.
- Line 164: "provide clear evidence of this idea" Which evidence? Which idea?
- Line 169: "...provides laboratory evidence that suggests..." Sounds strange. Maybe "...provides evidence for the suggestion..."?
- Line 169: Maybe the authors should state here the findings/evidence of their study?
- Line 179: exchange "never" with "unlikely"
- Line 204-205: already mentioned in line 175-177
- The authors should include their recent publication that analyze the α COP gene as RNAi target for *B. aeneus* in a different rape bud assay (Willow et al. 2020) and discuss the improvement of the bud assay. In Willow et al. 2020 feeding of 5 μ g/ μ l ds α COP showed only 16 % mortality, compared to >75 % at d15 in the present study using the same dsRNA concentration that targets the same gene, but using anthers instead of green buds.

In general, I would suggest to discuss the results of this study in comparison to other SIGS approaches that have been performed in other insects in more detail. The mortality presented in this study is relatively moderate compared to other studies on leaf eating insects like CPB. Work from Máximo and colleagues (2020) showed much higher mortality rates with lower dsRNA concentrations, to name just one. However, including more information about new formulation would be a benefit for the manuscript, showing possible options to overcome these obstacles.

Methods

- No headings?
- Line 225-229: merge first sentence with third and exchange "our control" with "dsRNA control" and "our target gene" with "RNAi target gene", if needed at all. Maybe "dsRNA control" after gfp is sufficient

- Line 237 + 250: "Eight randomly chosen... beetles" and "n=15" For which treatment did you use eight beetles? I thought you used 15 animals for each treatments?
- Line 242-243: delete "There were"
- Line 269: exchange "Quantistudio" with "QuantStudio"
- Line 280-284: Is there any difference between these two sentences?

References

- Only some references have the DOI number attached, for consistency I recommend to either add all or delete the once included

Recommended references:

Knorr E, E. Fishilevich, L. Bingsohn, M. Frey, M. Rangasamy, A. Billion, S. Worden, P. Gandra, K. Arora, W. Lo, G. Schulenberg, P. Valverde, A. Vilcinskas & K. E. Narva 2018: Gene silencing in *Tribolium castaneum* as a tool for the targeted identification of candidate RNAi targets in crop pests. *Scientific Reports* 8:2061. DOI:10.1038/s41598-018-20416-y

Máximo, Wesley P. F.; Howell, Jeffrey L.; Mogilicherla, Kanakachari; Basij, Moslem; Chereddy, Shankar C. R. R.; Palli, Subba R. (2020): Inhibitor of apoptosis is an effective target gene for RNAi-mediated control of Colorado potato beetle, *Leptinotarsa decemlineata*. In: *Archives of insect biochemistry and physiology* 104 (4). DOI: 10.1002/arch.21685.

Willow, Jonathan; Soonvald, Liina; Sulg, Silva; Kaasik, Riina; Silva, Ana Isabel; Taning, Clauvis Nji Tizi et al. (2020b): First Evidence of Bud Feeding-Induced RNAi in a Crop Pest via Exogenous Application of dsRNA. In: *Insects* 11 (11), S. 769. DOI: 10.3390/insects11110769.

We are very grateful to both reviewers for providing us their expertise and constructive comments on our manuscript. Below are the reviewers' comments in **Bold** text, each followed by our response.

Reviewer 1

Comment:

My greatest concern is that this use case as presented would involve treating a flowering crop, probably repeatedly, to target pollen beetles feeding on open flowers. This by itself would only help to reduce the population of pollen beetles for the next season and would not help to reduce economic damage to the crop, which is caused mainly by feeding damage to closed buds before the flowering phase; during the flowering phase the beetles are not considered a pest. I feel this approach may not be very popular with farmers! The dsRNA technology would therefore be better sprayed at the bud stage, which is the damage-susceptible stage to this pest, to prevent feeding damage on buds - as in the authors' previous work which is not mentioned (Willow et al. 2020 First Evidence of Bud Feeding-Induced RNAi in a Crop Pest via Exogenous Application of dsRNA. *Insects* 11.11:769). This needs to be acknowledged. However I can envisage a practical and successful use case for the SIGS or HIGS approach via feeding in open flowers as part of a more integrated trap cropping strategy (i.e. where an early flowering trap crop is treated (SIGS) or used (HIGS) to prevent infestation and damage of the main cash crop during its damage-susceptible bud stage). This would have the added advantage that only a small area is treated and so would be economically more viable than treating the whole crop.

Response:

Thanks for this great idea, to link RNAi with trap cropping. We have included this concept into both our introduction (lines 61-63, 116-117) and discussion (lines 186-188). We also now mention our recently-published bud feeding RNAi paper in both introduction (lines 109-111) and discussion (lines 199-205). Our apologies that the latter paper was still under construction when we submitted the present manuscript.

Comment:

Pollen is not critical to the survival of pollen beetle (see Cook et al 2004 Do pollen beetles need pollen? The effect of pollen on oviposition, survival, and development of a flower-feeding herbivore. *Ecological Entomology* 29 (2):164-173; and Corda et al (2018) Impact of flower rewards on phytophagous insects: importance of pollen and nectar for the development of the pollen beetle (*Brassicogethes aeneus*). *Arthropod-Plant Interactions* 12(6):779-785). This phrasing therefore needs to be modified but moreover, this raises the issue that if pollen beetles can detect treated anthers and prefer untreated material, they could switch to feeding on other floral parts including nectar, which would decrease efficacy of RNAi via anther feeding. This supports the potential inclusion of nectar-mediated RNAi technology in the crop protection strategy as suggested in their previous work (Willow et al submitted). I feel, however, that if pollen beetles can detect and avoid treated anthers only a HIGS approach could be successful as untreated anthers would be common in a SIGS approach with new flowers appearing daily - repeated spraying could therefore be ineffective and/or impractical (and would also increase the carbon footprint of the crop production strategy).

Response:

Thanks for this clarification. We have now included a statement on the pollen beetle's level of pollen requirement, with references (lines 114-115). While we have never conducted choice-tests between untreated and dsRNA-treated food sources, we have always observed high amounts of dsRNA-treated food consumption in honey-solution-, bud- and anther feeding experiments (e.g. in honey-solution tests, we dyed the solution blue to confirm feeding); and dsRNA-treated buds and anthers always showed clear indications of consumption. Any avoidance of dsRNA-treated material would indeed suggest that a HIGS approach could be more effective. However, all evidence gathered by us so far

suggests that the pollen beetles do not avoid dsRNA. This may be due to it being odourless. This type of choice experiment would indeed be useful down the line.

Comment:

The manuscript would benefit from certain terms being better defined to help the non-specialist reader:

- 1. How does double stranded RNA work (briefly) in terms of producing RNA interference effect? (Introduction)**
- 2. what is α COP (ds α COP) (Abstract & introduction) and what does the α COP protein do to make it vital? (L88)**
- 3. bp should be written in full for first use in Methods section (L224)**

Response:

Thank you. We have added a brief explanation of how dsRNA produces the RNAi effect (lines 71-75), as well as an overview of the role of alphaCOP in insects (lines 101-107). Also, we defined bp at first mention (line 259).

Comment:

Some clarifications to the M&M would be helpful:

- 1. Abstract - duration of short-term exposure (3d) given but not term of chronic exposure (17d) ;**
- 2. L239 Were anthers from treated flowers of various ages or from flowers of a standard age? was the pollen dehisced or not or a mixture of both?**
- 3. L241 How long was treatment vortexed for?**
- 4. L103 & 243 – It is unclear whether anthers for the short term treatment were provided with treated anthers which were left for 3d or if anthers were changed daily as with chronic treatment anthers (Fig 1 suggest the latter) -needs to be made clear in the text as well as the duration of the chronic exposure (= experimental duration 17d) – this should come sooner than L250.**
- 5. Why was experimental duration 17d and not, say 14 or 21 days?**

Response:

Chronic exposure is now described as 17 d in abstract (line 31).

Anthers were standardised, in that we always used flowers at the similar stage of being both dehisced and fresh, and we have now included this detail (lines 277-278).

Treatments were vortexed for 10 s (line 280).

We now make it clear that all anthers for all treatments were changed and treated on the day of their provision (line 277).

Experimental duration was 17 d because, after this time, flower availability for our number of samples became less reliable. We decided to go further than 14 d, due to the delayed action of dsRNA.

Comment:

- 1. In the first line of the abstract ‘represents the most biosafe insecticidal compounds’ - represent should be singular? Also, there are two ‘biosafe’ words in the same sentence. Last sentence of the Abstract also needs attention (two approaches, clunky split)**

Response:

We pluralised dsRNA (line 29) to make the sentence more correct.

We elaborated on the next steps for researching both approaches (SIGS and HIGS), and tend to each one separately (lines 35-40).

Comment:

L64-67 sentence is clunky with use of two semicolons

Response:

We removed semicolons to create separate sentences, making it easier to read (lines 75-79).

Comment:

L83 as these acquire

Response:

We now changed it to as you suggested (line 94).

Comment:

L89 (Willow et al. In Press) should come at the end of the sentence

Response:

We now placed the citation at the end of the statement (line 109), and changed the citation as it has since been published.

Comment:

L81 two uses of ‘successive’ in this sentence

Response:

We have altered the text to remove repetitive wording (line 93).

Comment:

L83 acquire ; L96 & 105 short term not shortterm; L106 set-up not setup

Response:

We have changed to “acquire” (line 94).

We use to hyphenated “short-term” in both instances, as this is the correct way to write it (lines 121 and 123).

As both “set-up” and “setup” are correct as used often in both US and UK English, we maintain our use of “setup”, as this is our preference. We hope the reviewer can understand and agree to this.

Comment:

L231 as pollen is not critical for survival as mentioned above, please also delete ‘on pollen of’

Response:

We removed this. It now reads “allowed to feed ad libitum on oilseed rape flowers” (line 269).

Comment:

L232 crop not field - the field contains the crop!; Eight fast moving beetles (used as a proxy...) were selected at random and introduced...

Response:

We changed “field” to “crop” (line 268).

We changed the beetle selection statement to as you suggested (line 274).

Comment:

L247 wettened? Past tense of wet is wetted – was it saturated or moistened? these would possibly be better terms

Response:

The cotton was saturated, so we changed to this term (line 286).

Comment:

L155 no comma

Response:

We removed this part of the paragraph, where we were repeating results.

Comment

L159 replace ‘as that which would’ to ‘compared to that achieved’ L160 a higher concentration or higher concentrations

Response:

We changed our “achieved” wording to what you suggested (lines 179-180).

We changed this to “higher dsRNA concentrations” (line 180).

Comment:

L164 evidence to support (not of) this idea

Response:

We changed this to “... evidence to support this idea” (line 184-185).

Comment:

L184 anthophilous

L186 delete larvae after instar;

L188 insert stages before simultaneously

Response:

We corrected the anthophilous spelling (line 212).

We deleted “larvae” after “instar” (line 221).

We inserted “stages” before “simultaneously” (line 223)

Comment:

L192 rephrase ‘conceivably touch upon the concept of HIGS vs SIGS are the most optimal’ – it’s a question not a concept?

Response:

We rephrased this to “raise the question”, rather than “conceivably touch upon” (line 227).

Comment:

L200 development of an RNAi cultivar or development of RNAi cultivars

L209 target specific.

Response:

We removed “an” before “RNAi cultivars” (line 235).

We maintain “target-specific” as a hyphenated word, as “target-specific” is an adjective here (lines 243-244).

Comment:

Several references are needed:

1. L66 the first strategy needs a reference?;

2. L230 reference for use of gel electrophoresis for this purpose?;

3. L234 reference for species determination of *B. aeneus* from other related species;

4. L255 reference for relative gene expression analysis via qPCR? 5. When names are cited in the text these seem not to be numbered as with other citations e.g. Mauchline et al 2018 (L51; #1) , Cagliari et al 2018 (L67; #14) are missing citation numbers ; Willow in press should be ref #20 and should be cited (L89, 169)?

Response:

The HIGS approach is referenced in line 88 (reference 17). We have now also added reference a review that touches on different approaches (line 78).

Gel electrophoresis is the standard “tool” used to make sure that the only bands you detect in your sample are those corresponding to the length of your experimental dsRNA. Perhaps an analogy to citing this method would be citing the use of a microscope to identify a cell type. We hope the reviewer can understand why we do not reference anything for our use of gel electrophoresis.

We now reference the literature we used to distinguish *B. aeneus* from other pollen beetle species (lines 269-270).

We do not reference the use of qPCR for relative gene expression, as this represents the method by which all relative single-gene expression analyses are performed. We hope the reviewer can understand why we do not reference anything for our use of this method.

Mauchline et al. and Cagliari et al. are now cited appropriately with citation numbers (lines 52 and 79). As Willow in press has now been published, it is cited accordingly (lines 109 and 192).

Comment:

Tables & Figures. Legends should be written in full such that all necessary information is presented to the reader to negate reference to the text. This related to all Figures and tables: 1. Table 1 - treatment comparisons are the same in each category, so I'm not sure this is really needed - could explain in the text or move to Supplementary material? ;

2. Fig 1 should describe what the 'experiment' was for (note set-up, not setup);

3. Figure 2. Describe 'each experiment' i.e. what were the treatments, rates, & brief description of what the test was. Is the % survival totalled over 3 reps or is it the mean % survival over 3 reps – what do the error bars represent SE mean or SE proportion? Key is difficult to interpret - different treatments (dsGFP and dsCOP) would be better just a coloured line with no circle as begs question about short-term feeding treatments (triangles). L403 Asterisk. Figs A-C are not strictly necessary - could Fig D be drawn on a finer scale to enable clearer separation between the different treatments?;

4. Fig 3 needs clearer description of the experiment i.e. results of gene expression to measure success of gene silencing, qPCR should be given in full legend, what are the units on the Y axis, explain what at 3d and 6d means i.e. from the start of the experiment/after feeding for 3d/6d on treated anthers.

Response:

Table 1 has been removed from main manuscript, and became Supplementary Table 3 (see Supplementary Material file), and is now cited as such (line 320).

Fig 1 is now Fig 3, and the appropriate changes have been made to the caption (lines 470-475).

Fig 2 is now Fig 1, and the caption now includes all necessary information to make things more clear (lines 452-464), as well as figure legends are now less confusing (use of coloured lines, rather than coloured lines with circles) and less cluttered (short-term and chronic symbols at the top of the figure, rather than in each box). See revised figure below.

Fig 3 is now Fig 2, and the caption now includes all necessary information to make things more clear (lines 464-468). The qPCR figure has no y-axis units because it shows relative gene expression, which consists of a ratio of how the target gene is expressed in the experimental samples compared to control samples.

Reviewer 2

Comment:

I recommend to include the pest name *Brassicogethes aeneus* and or pollen beetle, as this might increase the visibility of the manuscript

Response:

We changed “a crop pest” to “pollen beetle” (line 1).

Comment:

Line 29: “potentially represents” exchange with e.g. “ represents a promising ...”

Response:

We made this Exchange (line 29).

Comment:

Line 30: “...biosafe strategy for integrated management of the pollen beetle *Brassicogethes aeneus* is greatly needed ...” Why?

Response:

We have removed this part of the abstract due to word limit constraints.

Comment:

Line 32: "...ability to produce the RNA interference (RNAi) effect..." I suggest to use a different wording e.g. "induce" or "initiate" RNAi effects

Response:

We now use the word "induce" (line 30).

Comment:

Line 39-41: "Our results have implications for... as well as the need for research into the development and potential future use of host-induced gene silencing approaches." Is there something missing in this sentence?

Line 39: "Our results have implications for the development of RNAi strategies ..." Please state which implications.

Line 41: "the need for research into the development and potential future use of host-induced gene silencing approaches" Why?

Response:

Thank you. We now elaborated on our most important results in this statement, as well as why these results call for further investigation into HIGS approaches to pollen beetle management (lines 35-40).

Comment:

Line 66: "this control measure is potentially species-specific" Reference?

Response:

We have rephrased this sentence to reflect a more logically-thought-out, and conservative, perspective on target specificity (lines 69-71).

Comment:

Keywords - Exchange old name "Meligethes aeneus" with "Brassicogethes aeneus"

Response:

We now use *Brassicogethes aeneus* in the keywords instead (line 42).

We also removed both common names of the plant, and used only *Brassica* (line 42), and added "insect" (line 43).

Comment:

Line 53-62: This section seems to me a bit unstructured. Starting with "In order to achieve ecologically sustainable crop production...", but not mentioning why we have to achieve this. Next, the authors' states that an ecologically sustainable crop production should consists of "multiple approaches", but list only one approach: "enhancing biological control". This followed by the application of insecticide use. The authors should consider rewriting this section.

Response:

Thank you. We have added an introductory sentence to this paragraph, explaining why we must achieve ecologically sustainable crop production. We also added another approach for consideration in IPM, with appropriate reference to a review on the topic of trap cropping (lines 53-64).

Comment:

Line 89: The authors should consider citing the work of Knorr and colleges (2018) that showed functional RNAi in field collections of B. aeneus, as well as some information/references about the selected RNAi target gene

Response:

Thanks for this suggestion. We have added important information on the Knorr et al. 2018 paper, as well as information and references about our selected target gene (lines 99-107).

Comment:

Line 95: exchange “produce” with “induce”; add “of dsRNA” after “anther feeding”

Line 97-99: delete “Using gfp specific dsRNA (dsGFP) as a control” (Material & Methods)

Response:

We now use “induce”, and rephrased to “anther-based feeding of dsRNA” (lines 119-120).

We deleted gfp control from the paragraph.

Comment:

Line 98-100: At the end of the introduction you usually give a brief description of important results, and not the authors’ expectations.

Response:

We now added a brief description of important results, and deleted a statement on our expectations (lines 122-127).

Comment:

Line 103-107: Belongs to Material and Methods

Response:

We deleted this text from the results, and left it in the Material and Methods only.

Comment:

Line 155-157: too much details, belongs to Material and Methods. Instead the authors should start with re-stating the research problem they wanted to investigate

Response:

We removed the details that are already presented in the results, and added a sentence re-stating the research question (lines 177-178).

Comment:

Line 159: “similar effect as that which would be achieved from” sounds odd

Response:

We rephrased this to “similar effect compared to that achieved...” (lines 179-180).

Comment:

Line 161-164: I can’t follow the statement of this sentence.

Response:

We are referring to our evidence that suggests that lower concentrations can be applied successively in order to not only control pests better, but also potentially benefit agricultural practitioners financially, as lower concentrations are being used. We hope the review can understand what we mean here, and agree that it is written sufficiently in the manuscript.

Comment:

Line 164: “provide clear evidence of this idea” Which evidence? Which idea?

Response:

We have rephrased this as “... while we provide clear evidence to support this idea...” (line 184-185).

The idea we refer to is of that written in the precluding sentence. The evidence is that which we present in the results. We hope the reviewer can now find this to be sufficiently clear.

Comment:

Line 169: “...provides laboratory evidence that suggests...” Sounds strange. Maybe “...provides evidence for the suggestion...”?

Line 169: Maybe the authors should state here the findings/evidence of their study?

Response:

Thank you. We now elaborate on our findings in this sentence, and removed the word “laboratory” (lines 192-194).

Comment:

Line 179: exchange “never” with “unlikely”

Response:

We have now exchanged these words (line 207).

Comment:

Line 204-205: already mentioned in line 175-177

Response:

We have deleted this statement from the earlier paragraph, and now mention it only in lines 237-238.

Comment:

The authors should include their recent publication that analyze the α COP gene as RNAi target for *B. aeneus* in a different rape bud assay (Willow et al. 2020) and discuss the improvement of the bud assay. In Willow et al. 2020 feeding of 5 μ g/ μ l ds α COP showed only 16 % mortality, compared to >75 % at d15 in the present study using the same dsRNA concentration that targets the same gene, but using anthers instead of green buds.

In general, I would suggest to discuss the results of this study in comparison to other SIGS approaches that have been performed in other insects in more detail. The mortality presented in this study is relatively moderate compared to other studies on leaf eating insects like CPB. Work from Máximo and colleagues (2020) showed much higher mortality rates with lower dsRNA concentrations, to name just one. However, including more information about new formulation would be a benefit for the manuscript, showing possible options to overcome these obstacles.

Response:

Thank you for these suggestions. We have added information about our previous honey-solution and bud assays, and with possible explanations on why we see these differences in RNAi efficacy between studies (lines 199-206).

We also now acknowledge the moderate nature of RNAi efficacy in *B. aeneus*, and discuss how several other coleopteran pests exhibit more robust RNAi via feeding at lower doses of target-specific dsRNA. We included references to numerous relevant studies (lines 213-219).

Comment:

Methods

No headings?

Line 225-229: merge first sentence with third and exchange “our control” with “dsRNA control” and “our target gene” with “RNAi target gene”, if needed at all. Maybe “dsRNA control” after *gfp* is sufficient

Line 237 + 250: “Eight randomly chosen... beetles” and “n=15” For which treatment did you use eight beetles? I thought you used 15 animals for each treatments?

Line 242-243: delete “There were”

Line 269: exchange “Quantistudio” with “QuantStudio”

Line 280-284: Is there any difference between these two sentences?

Response:

Thank you. Headings are now added to Methods section (lines 259, 267, 272 and 320).

We connected the first and third sentences of the dsRNA section in the methods, and removed unnecessary wordings as you suggested (lines 259-263).

Eight beetles were introduced to each cage, and we had 15 cages (n here represents the sample size, and equals 15 cages). We have added the word “cages” to the sample size statement to make this more clear (line 287).

We maintain the words “There were” before saying “eight treatments”, as we must keep this in the text to maintain our statement as a complete sentence. We hope the reviewer agrees to this.

We changed Quantistudio to QuantStudio (line 308).

Each sentence introducing the statistics section is necessary, as we must explain what is shown in Supplementary Table 3 (previously Table 1), that being what was compared within short-term and chronic exposure groups, as well as what was compared between short-term and chronic exposure groups. We hope the reviewer can agree that we explain this in the text in a sufficiently concise manner. Additionally, we found a mistake in the text, and fixed it in order to indicate that gene silencing was not statistically compared between short-term and chronic feeding groups (text is now consistent with Supplementary Table 3 (lines 320-327)).

Comment:

Only some references have the DOI number attached, for consistency I recommend to either add all or delete the once included

Recommended references:

Knorr E, E. Fishilevich, L. Bingsohn, M. Frey, M. Rangasamy, A. Billion, S. Worden, P. Gandra, K. Arora, W. Lo, G. Schulenberg, P. Valverde, A. Vilcinskas & K. E. Narva 2018: Gene silencing in *Tribolium castaneum* as a tool for the targeted identification of candidate RNAi targets in crop pests. *Scientific Reports* 8:2061. DOI:10.1038/s41598-018-20416-y

Máximo, Wesley P. F.; Howell, Jeffrey L.; Mogilicherla, Kanakachari; Basij, Moslem; Chereddy, Shankar C. R. R.; Palli, Subba R. (2020): Inhibitor of apoptosis is an effective target gene for RNAi-mediated control of Colorado potato beetle, *Leptinotarsa decemlineata*. In: *Archives of insect biochemistry and physiology* 104 (4). DOI: 10.1002/arch.21685.

Willow, Jonathan; Soonvald, Liina; Sulg, Silva; Kaasik, Riina; Silva, Ana Isabel; Taning, Clauvis Nji Tizi et al. (2020b): First Evidence of Bud Feeding-Induced RNAi in a Crop Pest via Exogenous Application of dsRNA. In: *Insects* 11 (11), S. 769. DOI: 10.3390/insects11110769.

Response:

Great suggestions. Your suggested references have been implemented in our paper, among several others we now believed to be necessary.

DOIs have been removed for consistency across references.

Other changes to the manuscript:

As Communications Biology guidelines request that abstracts are approximately 150 words or less, we have decreased the word count of our abstract to 169 words, leaving in only what we believe are the most necessary statements and details. We hope that the reviewers can find our changes to the abstract acceptable.

REVIEWERS' COMMENTS:

Reviewer #1 (Remarks to the Author):

The concept of RNAi is still relatively new and offers exciting possibilities for targeted control of pests such as *Brassicoglyphus aeneus*, an important flower-feeding pest of oilseed rape. The work presented here showed that dsRNA (dsCOP) applied to anthers induced RNA interference and reduced survival, particularly following chronic exposure suggesting strong potential for use in biosafe management strategies for this pest. The practical work in this study is novel, has been very well conducted, statistics are appropriate, and the manuscript is well written. The authors have addressed all of my comments from the previous version. One small additional suggestion is that as well as trap cropping, an obvious pest management tool not included in the IPM section in the Introduction (Lines L53-67) is development/use of pest resistant lines - as the work presented here ultimately suggests development of RNAi cultivars would be useful. Mention of this IPM tool would make this section more complete e.g reviews by Hervé <https://doi.org/10.1007/s11829-016-9438-8> ; <https://doi.org/10.1111/pbr.12552>

I believe this work will be well received by academic and industrial stakeholders involved in environmentally sustainable pest management.

A few minor typos:

Line 7 – should it be Department of plant health or school of plant health? 'Chair' usually represents the professorial position of a person

148 Delete 'These include' as you indicate there are 2 strategies – so should describe these two ('these include' doesn't fit with the definition that there are two)?

247-8 insert a (compared to a single...)

369 Indeed is superfluous – suggest deleting it.

374 Should give full genus name if presented at the start of a sentence or paragraph.

374 suggest replacing reproductive with flower? Do plants produce flowers for anything other than reproduction?

374-376 a citation should be provided to support these statements?

388 case by case repetition of line 386

Reviewer #2 (Remarks to the Author):

The authors did a very good job in editing the manuscript and addressed all remarks to my complete contentment. I do not have any further objections and recommend the manuscript for publication.

Responses to reviewer 1:

Comment:

One small additional suggestion is that as well as trap cropping, an obvious pest management tool not included in the IPM section in the Introduction (Lines L53-67) is development/use of pest resistant lines - as the work presented here ultimately suggests development of RNAi cultivars would be useful. Mention of this IPM tool would make this section more complete e.g reviews by

Hervé <https://doi.org/10.1007/s11829-016-9438-8> ; <https://doi.org/10.1111/pbr.12552>

Response:

We now include a statement about this (lines 64-67), and cite the overarching review on this topic (Hervé 2018), one of the references suggested by reviewer 1.

Comment:

A few minor typos:

Line 7 – should it be Department of plant health or school of plant health? ‘Chair’ usually represents the professorial position of a person

148 Delete ‘These include’ as you indicate there are 2 strategies – so should describe these two (‘these include’ doesn’t fit with the definition that there are two)?

247-8 insert a (compared to a single...)

369 Indeed is superfluous – suggest deleting it.

374 Should give full genus name if presented at the start of a sentence or paragraph.

374 suggest replacing reproductive with flower? Do plants produce flowers for anything other than reproduction?

374-376 a citation should be provided to support these statements?

388 case by case repetition of line 386

Response:

‘Chair of Plant Health’ is the name of our department.

We fixed the wording regarding the two strategies, and made the sentence technically correct (lines 80-81).

We changed to ‘compared to a single high-concentration treatment’ (lines 129-130).

We deleted ‘Indeed’ where it was indicated to be unnecessary.

We changed ‘*B. aeneus*’ to ‘Pollen beetle’; and we changed ‘reproductive’ to ‘flower’ (line 225).

We now include a citation for this statement on larval pollen beetle ecology (line 227).

We removed, ‘though this must also be taken on a case by case basis’, making it now read better.